# Hepatic transcriptome profiling unveils candidate genes in cattle with liver abscesses under the influence of beef genetics in dairy cattle

Luana D. Felizari[1], Sydney M. Bowman[1], Chiquito J. Crasto[2], Jhones O. Sarturi[1], Dale R. Woerner[1], Bradley J. Johnson[1]*

**1** Department of Animal and Food Sciences, Texas Tech University, Lubbock, Texas, United States of America, **2** Center for Biotechnology and Genomics, Texas Tech University, Lubbock, Texas, United States of America

* bradley.johnson@ttu.edu

## Abstract

Liver abscesses are a significant concern in cattle feeding, linked to visceral condemnation and carcass trimming; however, the molecular mechanism of development and progression of liver abscesses is unknown. This study aimed to evaluate the hepatic transcriptomic profile, immunohistochemistry, and IGF-I circulation in beef × dairy (Angus × Holstein) steers with and without liver abscesses. Samples were collected from twelve steers (final body weight of 719 ± 5.8 kg) originating from the same feedlot and were selected based on liver scores at harvest. The animals were divided into abscessed ($n=6$) and healthy livers ($n=6$). Blood samples were used to measure circulating insulin-like growth factor I (IGF-I) levels using an ELISA kit. Liver samples were divided into two portions; one portion was used for immunohistochemistry (IHC) to identify IGF-I receptor (IGF-IR) abundance, while the second portion was used for RNA extraction, library preparation, and sequencing (Illumina NovaSeq 6000 platform). Differentially expressed genes (DEGs) were identified with the DESeq2 R package, using an adjusted $p$-value ≤ 0.05 and fold change > 1.5. Sera IGF-I was not affected by liver condition; however, IGF-IR abundance was up-regulated in abscessed livers. A total of 568 DEGs were identified, with 372 up-regulated and 196 down-regulated in abscessed livers. Notably, the most highly up-regulated genes were *FGF23*, *NXPH4*, and *CYP7A1*, while *EPHA6*, *CD70*, and *INHBA* showed the most significant downregulation. Protein-protein interaction (PPI) network analysis identified *THBS1* and *COL1A2* as significant hub genes. The DEGs showed enrichment in biological processes related to angiogenesis, cell migration, adhesion, and extracellular matrix organization. Pathway analysis indicated activation in signaling pathways, including hepatic fibrosis, interleukin, and IGF-I signaling. These findings reveal candidate genes and pathways linked to inflammatory responses and tissue remodeling, offering valuable evidence that enhances our understanding of the progression of liver abscesses in cattle.

**Data availability statement:** All relevant data are within the paper and its Supporting Information files.

**Funding:** This study was funded in part by the Gordon W. Davis Regents Chair (to BJJ). No additional external funding was received for this study. The funders had no role in study design, data collection, and analysis, decision to publish, or preparation of the manuscript

**Competing interests:** The authors have declared that no competing interests exist.

## Introduction

Liver abscesses represent a meaningful economic challenge in the cattle industry, adversely impacting feedlot operations and packing plants. The prevalence of liver abscesses ranges from 12% to 32% in feedlot cattle [1]. Such findings encumber financial losses exceeding $41.5 million annually due to liver abscess condemnation [2]. Holstein steers offered high-grain diets exhibited, on average, a greater incidence of liver abscesses compared to beef breeds or heifers, 25%, 18.2%, and 19.1%, respectively [2]. In addition, [3] observed liver abscess rated as high as 50% in straightbred dairy cattle; however, the reason for the elevated prevalence of liver abscesses in dairy-type is still unclear.

The principal pathogen linked to liver abscesses in cattle is *Fusobacterium necrophorum*, commonly found in the rumen and gastrointestinal tract [4]. The pathogenesis of liver abscesses typically involves the translocation of *F. necrophorum* from the rumen to the liver via the portal vein, potentially causing infection and liver abscess formation. Notably, a study has demonstrated that regardless of the presence of *F. necrophorum* in the lungs and spleen following *F. necrophorum* systemic injection in mice, abscesses develop exclusively in the liver [5], suggesting that the liver has unique molecular and immunological responses that facilitate abscess formation.

Insulin-like growth factor I (IGF-I) is primarily produced in the liver, and the concentration level in the blood can be influenced by cattle type and steroid hormones [6–8]. Studies have shown that dairy-influence cattle are associated with higher circulating IGF-I levels than straightbred beef cattle [9]. [10] reported a significantly higher concentration of IGF-I levels in Holstein steers, followed by dairy-type steers, and then Angus steers. The same authors reported that implanted steers had higher IGF-I concentrations than those not implanted, which may be explained by dairy influence, possibly more sensitive to steroid hormones than beef breeds [9]. Despite these findings, the relationship between IGF-I levels and liver abscesses remains unclear. [11] reported that IGF-I levels in the blood did not differ between animals with and without liver abscesses. However, more studies are needed to understand the relationship between IGF-I level and liver abscess.

Studies have focused on the impact of liver abscesses on performance and welfare, as well as strategies to identify and mitigate liver abscesses [12–18], while few have directly compared the molecular changes in tissues of animals with and without liver abscesses. Also, some studies have explored differences in the rumen [19] and ileal epithelium of animals with or without liver abscesses [20]. However, there are a lack of studies specifically investigating the liver tissue, which is crucial for understanding the local molecular environment supporting abscess formation. The objective of this study was to evaluate the hepatic transcriptomic profile, immunohistochemistry, and IGF-I circulation from beef × dairy (Angus × Holstein) with and without liver abscesses.

## Materials and methods

### Animal care and use

The following experiment was a collaborative study conducted at a commercial cattle feeding facility. All research followed the guidelines stated in the *Guide for Care and*

*Use of Agricultural Animals in Agricultural Research and Teaching* [21]. Researchers from Texas Tech University were involved only with postmortem sample collection, sample analysis, and statistical analysis.

## Animals and sample collection

The animals used in this experiment were obtained through artificial insemination, in which semen from beef bulls was used to inseminate dairy cows, resulting in twenty-nine beef × dairy (Angus × Holstein) crossbred progeny born between 06/2022–08/2022. Subsequently, the calves were raised at the same calf ranch until they reached an average weight of approximately 250 kg. Following this, animals were transported to a commercial feedlot, where they were assigned to the same pen until reaching 288 days on feed. During the feedlot time, the cattle were fed a diet consisting of 69.4% steam-flaked corn and 7.8% roughage, with net energy for gain ($NE_g$) of 1.52 Mcal/kg. The animals were implanted on the first day upon arrival (Synovex Choice, Zoetis, Parsippany, NJ) and reimplanted after 92 days on feed (Synovex One Feedlot, Zoetis, Parsippany, NJ).

After the cattle reached 288 days on feed, a total of twenty-nine steers were harvested at a commercial packing plant located in Dodge City, Kansas. Briefly, animals were stunned using a penetrating captive bolt prior to exsanguination, in accordance with the American Veterinary Medical Association (AVMA) Guidelines for the Euthanasia of Animals [22]. Anesthesia or analgesia was not administered before harvest, as the stunning method ensured immediate insensibility. Cattle had free access to water while being held before harvest, and all efforts were made to minimize stress and ensure animal welfare throughout the process. Livers were scored and collected using the Eli Lilly Liver Check System (Elanco, Greenfield, IN). Liver scoring was evaluated on a scale of 0 (no abscesses), A- (1–2 small abscesses), A (2–4 small active abscesses), A+ (1 or more large active abscesses), and A+AD (liver adhered to the gastrointestinal tract) by a trained person. Among the 29 total liver samples, the scores were: 51.72% scored 0, 6.9% scored A-, 13.79% scored A, 17.24% scored A+, and 10.35% scored A+AD. Subsequently, 12 A×H steers (BW = 719 ± 5.8 kg) were selected for the liver collection and divided into abscessed livers (*n* = 6, score = A-, A, A+), and healthy livers (*n* = 6, score = 0), considering the animal as the experimental unit.

After stunning, blood samples were collected from all steers using 50 mL conical centrifuge tubes (ThermoFisher) during the bleeding process. The tubes were stored on ice during transport and then refrigerated for 24 hours before centrifugation. Liver samples of each animal (*n* = 12) were collected immediately after slaughter at the same anatomical location on the right lobe. To avoid potential effects on our analysis, the collected liver tissue samples did not contain visible abscesses. Approximately 300 mg of tissue designated for gene expression were placed in 2 mL RNase-free microcentrifuge tubes (Invitrogen; Thermo Fisher Scientific, Inc.) containing 1.5 mL of RNA-Later (Thermo Fisher Scientific, Inc.) and placed on dry ice. Duplicate tubes were prepared for each sample. With the aid of a scalpel, portions of rectangular size (2.54 by 1.27 cm) were obtained to be used for immunohistochemical analyses. The samples were placed into a mold, embedded in clear frozen section compound (VWR International, West Chester, PA, USA), and frozen using 2-methylbutane cooled with dry ice, following a protocol adapted from [23]. Subsequently, liver samples were transported to Texas Tech University in a cooler with dry ice for approximately 6 hours and stored at −80°C until further analysis.

## RNA isolation and purification

Each frozen liver tissue sample was pulverized into a fine powder in liquid nitrogen using a mortar and pestle. Subsequently, total RNA was extracted with TRIzol reagent (Invitrogen, USA) and further purified using RNeasy® Mini kit and RNase-free DNase Set (Qiagen, Germantown, MD) according to the manufacturer's instructions. The total RNA was then quantified with a Cytation™ 5 cell imaging multi-mode reader (BioTek Instruments, Winooski, VT), with an absorbance (A260/280) ratio between 1.8 and 2.0. RNA integrity was evaluated using the 5400 Fragment Analyzer (Agilent Technologies, Santa Clara, CA, USA). Samples with a RNA Integrity Number (RIN) averaging 8.3 (range 6.7 to 9.0) were selected for library preparation.

## RNA sequencing and data analysis

Library preparation and mRNA sequencing were performed by the Novogene Co., Ltd (CA, US). The library construction followed the Illumina instructions for RNA-seq, and the sequencing was performed using the Illumina NovaSeq 6000 platform (San Diego, CA, USA), generating high-quality 2 x 150 bp paired-end reads with approximately 30 million reads per sample. The quality of raw sequence data was assessed using FastQC (Version 0.11.5; https://www.bioinformatics.babraham.ac.uk/projects/fastqc/) to evaluate parameters such as per-base sequence quality, GC content, and duplication levels. Reads containing adapter sequences, poly-N, and low-quality bases were removed using Trimmomatic software (Version 0.39; [24]). The filtered sequences were aligned to the *Bos taurus* reference genome (UMD3.1; https://www.ncbi.nlm.nih.gov/datasets/genome/GCF_000003055.6/) using the Hisat2 (Version 2.0.5; [25]). The count of reads mapped to individual genes was determined using FeatureCounts (Version 1.5.0; [26]). Genes with low counts, as <10 total reads across all samples, were removed from the dataset. FPKM (Fragments Per Kilobase of transcript per Million mapped reads) values were calculated for each gene based on the length of the gene and the count of reads mapped to this gene [27].

## Differential expression analysis

The differential expression analysis was performed using the DESeq2 package [28] in R statistical software (Version 4.3.2) to identify differentially expressed genes (DEGs) of beef × dairy crossbreed, comparing abscessed to healthy livers. To control the false discovery rate, $p$-values were adjusted using the Benjamini-Hochberg method (FDR). The DEGs were considered significant with an adjusted $p$-value < 0.05 and fold change (FC) > 1.5 (log2fold change > 0.58). The DEGs results were categorized as either up-regulated or down-regulated based on their log2fold change values. Principal component analysis (PCA) and the heatmap were performed using the ggplot2 (Version 3.5.1; [29]) and pheatmap (Version 1.0.12; [30]) packages in R, respectively. Volcano plots were visualized using VolcaNoseR web tool (https://github.com/JoachimGoedhart/VolcaNoseR; [31]).

## Functional enrichment analysis

Database for Annotation, Visualization, and Integrated Discovery (DAVID; https://david.ncifcrf.gov/) was used to enrich the Gene Ontology (GO; [32]) terms related to biological process (BP), molecular function (MF), and cellular component (CC) associated with the DEGs. *Bos taurus* annotation from the GO database and gene symbols within the list of DEGs were used to identify GO terms. The adjusted $p$-value ≤ 0.05 was considered significantly enriched.

The enriched canonical pathway was performed using Ingenuity Pathway Analysis (IPA; QIAGEN Redwood City, CA, USA; www.qiagen.com/ingenuity) based on DEGs datasets [33]. The core analysis was constructed by inputting the gene IDs and Log2FC values from the DEGs datasets. Fisher's exact test was employed to compare the overlaps between DEGs datasets and the IPA knowledge database, identifying significant canonical pathways with a $p$-value ≤ 0.05. The Z-scores calculated on IPA were used to determine the activation (Z-scores ≥ 2) or inhibition (Z-scores ≤ −2) of each canonical pathway.

## Analysis of the protein-protein interaction (PPI) network

To identify the PPI network, we uploaded the DEGs into the Search Tool for the Retrieval of Interacting Genes (STRING) database (http://string-db.org/). CytoHubba (Version 0.1) plugin in Cytoscape (Version 3.10.2; https://cytoscape.org/) was used to identify the hub genes using the degree method [34].

## Serum analysis

After the collection, tubes were transported and stored at 4°C for 24 hours to allow clotting. Subsequently, tubes were centrifuged at 1,250 × g at 4°C for 25 min, and the serum samples were aliquoted in 1.5 mL volumes for storage at −80°C.

Sera ($n = 12$) were analyzed to quantify the circulating concentration of insulin-like growth factor I (IGF-I). The concentration of circulating IGF-I was determined using a commercial ELISA kit (Quantikine Human IGF-I ELISA, R & D Systems, Minneapolis, MN) following the manufacturer's instructions and validated by [35]. Briefly, sera were 100-fold diluted with manufacturer supplied diluent, and standard curves ranged from 8 ng/mL to 0.125 ng/mL. Optical density readings were taken at 450 nm and 540 nm using an Epoch Microplate Spectrophotometer (BioTek), with background correction obtained by subtracting the 540 nm from the 450 nm (intra-assay CV: 3.5%).

### Immunohistochemistry

Embedded liver samples ($n = 12$) were transferred from −80°C to −20 °C for 24 hours. Samples were removed from the mold and sliced into 6-μm thick cross-sections at −20 °C with a cryostat (Leica CM1950; Leica Biosystems, Buffalo Grove, IL, USA). The sections were mounted on positively charged glass slides (Superfrost Plus, VWR International), with 3 sections per slide. The cross-sections were fixed in 4% paraformaldehyde in phosphate-buffered saline (PBS) (ThermoFisher) for 10 min at room temperature. Subsequently, the cross-sections were blocked with 0.2% Triton™ X-100 (ThermoFisher) in PBS, 2% bovine serum albumin (MD Biomedical, Solon, OH), and 5% horse serum (Invitrogen) for 30 min at room temperature. The cross-sections were incubated with primary antibody rabbit polyclonal anti-IGF-IR (1:200; Cat. No. ab182408, Abcam) at 4°C overnight. Subsequently, the slides were incubated with a secondary antibody for 45 min at room temperature in the dark (goat anti-rabbit, Alexa Fluor 488, Cat. No. A-11008; 1:1,000, Invitrogen). The slides were covered with thin glass coverslips (VWR International) with ProLong™ Gold Antifade Mountant with DAPI (Invitrogen). The samples were imaged in 3 random different spots (20X magnification) using the Cytation™ 5 cell imaging multi-mode reader (BioTek Instruments, Winooski, VT). The images were analyzed for total IGF-IR fluorescence using ImageJ software (ImageJ v1.53f51, NIH). The mean fluorescence intensity was calculated by averaging the values obtained from the images for each animal.

### Statistical analysis

The serum and immunohistochemistry data were analyzed by unpaired t-test using GraphPad Prism software version 10.2.3 (GraphPad Software, San Diego, CA, USA). The data were checked for normality using Shapiro-Wilk before applying for the statistical test. Statistical significance was considered at $p$-value ≤ 0.05.

## Results

### Illumina sequencing and read assembly

RNA-seq quality summaries and mapped reads from liver transcriptomes of beef × dairy steers under liver conditions as healthy and abscessed are presented in Table 1. On average, raw reads were approximately 71.4 million for abscessed livers and 65.0 million for healthy livers. After cleaning, the reads were reduced to 69.2 million and 63.3 million for abscessed and healthy livers, respectively. Additionally, on average, 91.3% of reads from abscessed livers and 92.2% of reads from healthy livers were uniquely mapped to the *Bos taurus* reference genome (UMD3.1) available on NCBI.

### Differentially expressed genes analysis

To identify DEGs, we employed the DESeq2 package in R, and a total of 16,506 genes were identified in the liver tissue comparing abscessed to healthy livers. Genes with *adjusted p-value* < 0.05 and Fold change > 1.5 were assigned as different expressions. The results showed that 568 genes were identified as DEGs, which included 372 genes up-regulated and 196 genes down-regulated in abscessed livers (S1 Table). The top 10 genes, up-regulated and down-regulated, are present in Table 2. The principal component analysis (PCA; Fig 1A) of DEGs between abscessed and healthy livers identified distinct clustering patterns. The PCA plot shows that PC1 and PC2 explain 24.06% and 13.86% of the variance,

**Table 1. Raw data, clean data, and quality from beef×dairy steers (*n*=12) with abscessed and healthy livers.**

| Item | Abscessed | Healthy |
|---|---|---|
| Raw reads | 71,483,646 | 65,073,393 |
| Clean reads | 69,217,417 | 63,306,499 |
| Total mapped | 65,046,568.17 (93.95%) | 60,042,562.50 (94.84%) |
| Multiple mapped | 1,900,682.17 (2.74%) | 1,691,982.33 (2.67%) |
| Uniquely mapped | 63,145,886.00 (91.21%) | 58,350,580.17 (92.17%) |
| Q20[1], % | 97.08 | 97.35 |
| Q30[1], % | 92.59 | 92.99 |
| GC[2], % | 49.46 | 49.15 |

[1]Percentage of bases with a Phred quality score of 20 and 30.

[2]Percentage of guanine (G) and cytosine (C) bases in the total nucleotide sequence.

**Table 2. The top 10 up-regulated and down-regulated differentially expressed genes (DEGs) between abscessed (*n*=6) and healthy (*n*=6) livers from 12 beef×dairy steers. Significant DEGs were defined as *adjusted p*-value<0.05 and fold change>1.5.**

| Up-regulated DEGs | | | Down-regulated DEGs | | |
|---|---|---|---|---|---|
| Gene | $Log_2FC^1$ | *p*-adj[2] | Gene | $Log_2FC^1$ | *p*-adj[2] |
| FGF23 | 5.424 | 0.00118 | EPHA6 | −4.409 | 0.01216 |
| NXPH4 | 5.177 | 0.00058 | CD70 | −2.162 | 0.00738 |
| CYP7A1 | 3.087 | 0.00601 | INHBA | −1.991 | 4.00E-07 |
| MSMB | 2.990 | 3.41E-06 | GRIN2C | −1.953 | 0.00397 |
| HSPA6 | 2.848 | 0.04218 | SULT1C3 | −1.800 | 0.00979 |
| GDF15 | 2.736 | 0.03492 | RXRG | −1.782 | 0.00812 |
| DLK1 | 2.730 | 0.00181 | SULT1C2 | −1.740 | 0.00001 |
| H2AC6 | 2.399 | 0.03204 | G0S2 | −1.727 | 0.01190 |
| CCDC201 | 2.210 | 0.00010 | ACACB | −1.717 | 0.02207 |
| HTRA3 | 2.180 | 0.00006 | HOPX | −1.703 | 0.01661 |

[1]$Log_2$ fold change represents the ratio expression levels comparing abscessed to healthy livers.

[2]*Adjusted p*-value (Benjamini-Hochberg false discovery rate).

respectively. Volcano plots were used to visualize the overall distribution of DEGs, with up-regulated genes shown in red, down-regulated genes in blue, and non-significant genes in grey (Fig 1B). Additionally, a heatmap was generated with the total DEGs, showing clear differences in expression levels between abscessed and healthy liver samples, with distinct clusters corresponding to up-regulated and down-regulated genes in red and blue, respectively (Fig 1C).

## Functional enrichment analysis of differentially expressed genes

The GO enrichment analysis identified 22 significant GO terms (*adjusted p-value* ≤ 0.05; S2 Table) in liver tissue when comparing abscessed to healthy livers. These significant terms were distributed into ten BP, eight CC, and four MF. Among the significant GO terms, angiogenesis (GO:0001525), collagen-containing extracellular matrix (GO:0062023), and heparin-binding (GO:0008201) were the most significant for molecular function, biological process, and cellular component, respectively (Fig 2). These GO terms indicate that differentially expressed genes play important roles in metabolic processes, extracellular matrix (ECM) composition, and molecular functions, potentially contributing to the progression of liver abscesses.

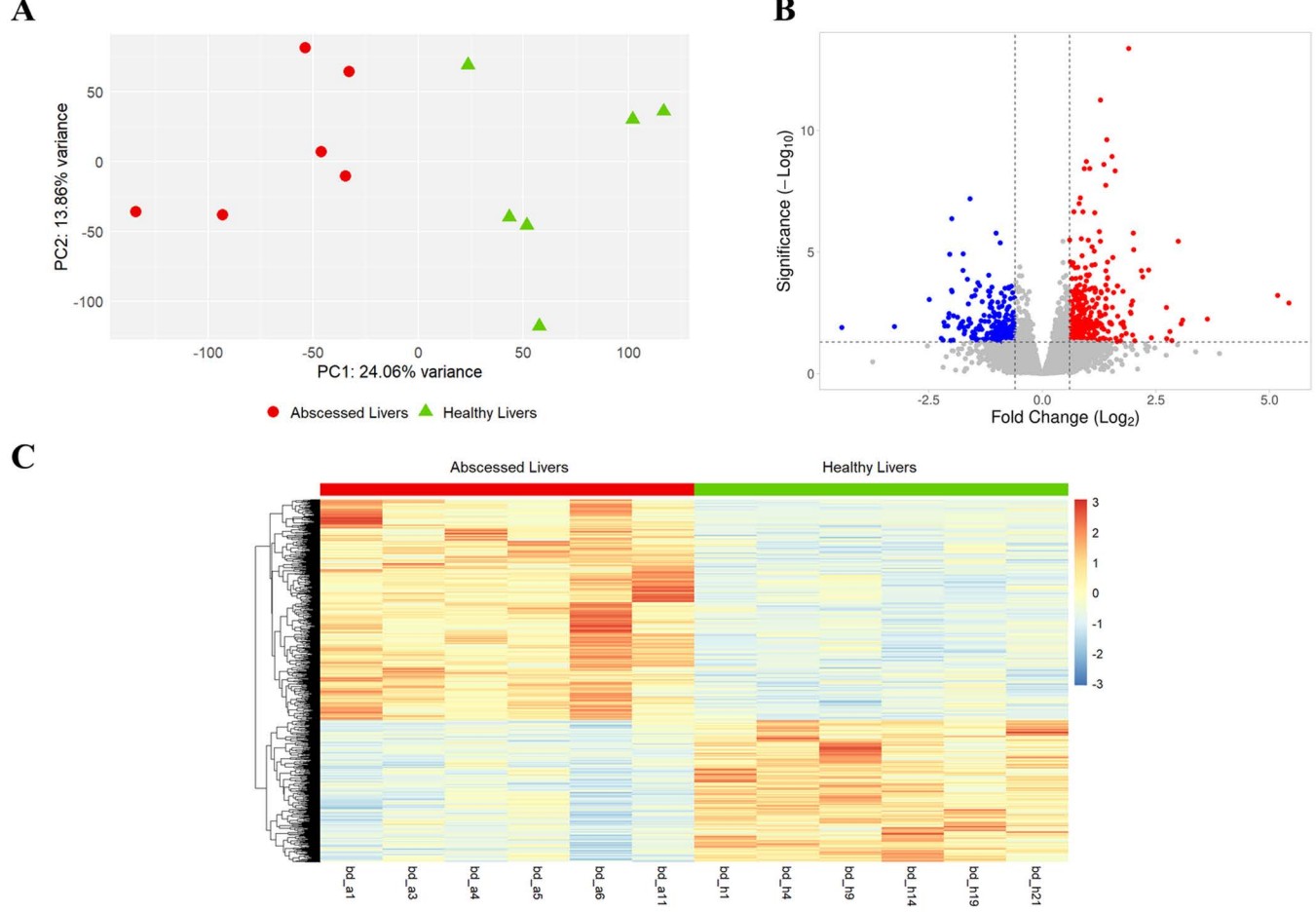

**Fig 1. Transcriptome analysis of liver tissue samples from 12 beef ×dairy steers, comparing abscessed ($n=6$) to healthy ($n=6$) livers.** (A) Principal Component Analysis (PCA) illustrating the separation along PC1 and PC2, with abscessed livers in red and healthy livers in green. (B) Volcano plot for DEGs comparing abscess to healthy liver tissue. The up-regulated genes (FC > 1.5 and *adjusted p*-value < 0.05) and down-regulated genes (FC < 1.5 and *adjusted p*-value < 0.05) are represented in red and blue dots, respectively. Dots with grey color represent the non-significant genes. (C) Heatmap displaying the expression levels of the DEGs in Log2FC, with healthy livers samples in green and abscessed livers samples in red. The columns and rows represent samples and genes, respectively.

The IPA software identified the top genes and canonical pathways associated with the DEGs. Table 2 lists the top 10 genes (up-regulated and down-regulated) in abscessed livers. Additionally, IPA identified 64 significantly enriched canonical pathways (S3 Table), with 60 activated pathways (Z-scores ≥ 2) and 4 inhibited pathways (Z-scores ≤ −2). Fig 3 shows the result of the selected pathways enriched associated with liver diseases, inflammation, proliferation, and cell death, ECM components, such as hepatic fibrosis signaling, extracellular matrix organization, interleukin-10 signaling, IGF-I signaling, NF-κB signaling, among others.

## Analysis of the PPI network

Based on the STRING database, interactions with a score greater than 0.6 were selected for analysis (S4 Table), resulting in the identification of 237 nodes (genes) and 415 edges (interactions). The identification of the most significant hub gene was *THBS1*, followed by *COL1A2*, *PTGS2*, *PECAM1*, *CCL2*, *ANGPT2*, *CXCL8*, *EDN1*, *COL4A*1, and *ICAM1*, as shown in Fig 4.

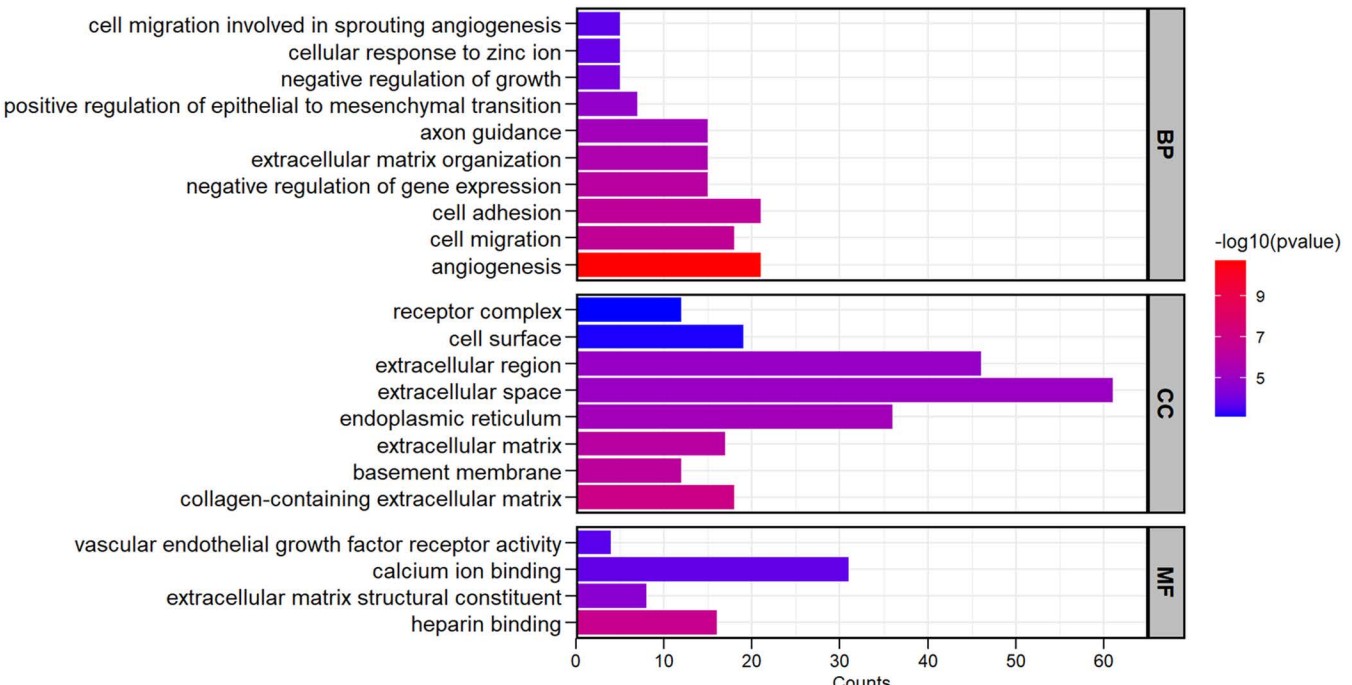

**Fig. 2. Enriched gene ontology (GO) terms of differentially expressed genes (DEGs) comparing abscessed (*n* = 6) to healthy (*n* = 6) liver tissues from 12 beef × dairy steers.** Significant DEGs were defined as adjusted *p*-value < 0.05 and fold change > 1.5. The GO terms are significantly enriched (*adjusted p*-value ≤ 0.05) and include terms related to biological process (BP), cellular component (CC), and molecular function (MF).

## Sera and immunohistochemistry

Sera analysis did not show any difference in IGF-I protein concentration levels between abscessed and healthy livers (*p*-value = 0.2244), as shown in Fig 5A (S1 Fig). Immunohistochemical images indicate an increased IGF-IR staining in abscessed livers with respect to healthy livers, as shown in Fig 5B. This result was confirmed by quantification analysis of the GFP signal, in which there was a significant (*p*-value = 0.0157) for higher IGF-IR staining in abscessed livers, as shown in the bar histogram in Fig 5C (S2 Fig).

## Discussion

Liver abscesses is a meaningful issue in the cattle feeding industry, with prevalence ranging from 12% to 32% of cattle [1]. Also, commercial packing plants have revealed instances of liver abscesses well over 50% in dairy-influenced cattle [3]. The mechanism of development and progression of liver abscesses is still unknown. To better understand the molecular mechanisms underlying this condition in beef × dairy steers, RNA-sequencing analysis, circulating IGF-I levels, and immunohistochemistry were conducted on liver tissue samples collected at the end of the finishing phase from steers with and without abscesses. In this study, 568 DEGs were identified when comparing abscessed and healthy livers, which include 372 genes up-regulated and 196 genes down-regulated. Among these, the ten most highly up-regulated genes were *FGF23*, *NXPH4*, *CYP7A1*, *MSMB*, *HSPA6*, *GDF-15*, *DLK1*, *H2AC6*, *CCDC201*, and *HTRA3*, while *EPHA6*, *CD70*, *INHBA*, *GRIN2C*, *SULT1C3*, *RXRG*, *SULT1C2*, *G0S2*, *ACACB*, and *HOPX* showed the most significant downregulation. For this study, we focused on the top three up-regulated genes, which are *FGF23*, *NXPH4*, and *CYP7A1*.

Fibroblast growth factor 23 (*FGF23*) is an endocrine hormone that is part of the FGF family and plays a crucial role in regulating phosphate and vitamin D metabolism [36]. *FGF23* is mainly produced from osteocytes in the bone; however,

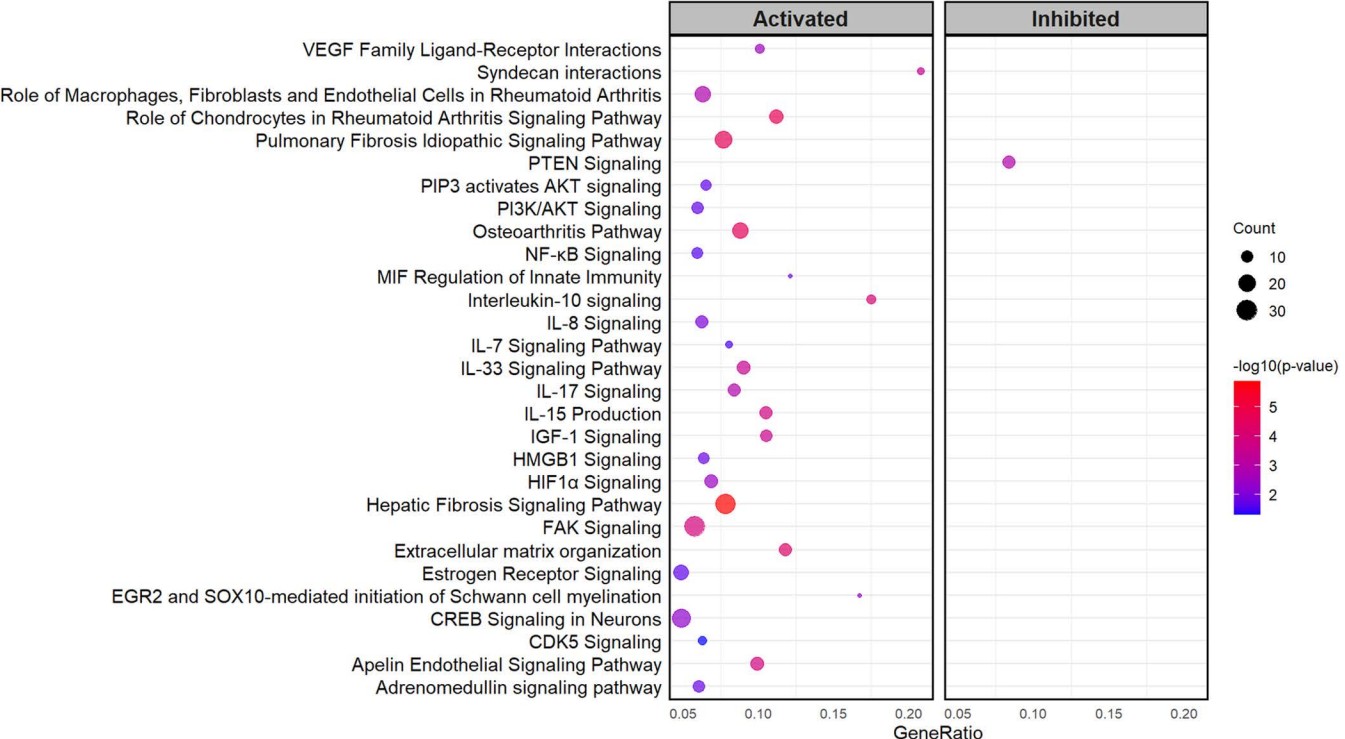

**Fig 3. Scatter plot of enriched canonical pathways (IPA) of differentially expressed genes (DEGs) comparing abscessed (*n* = 6) to healthy (*n* = 6) liver tissues from 12 beef × dairy steers.** Significant DEGs were defined as adjusted *p*-value < 0.05 and fold change > 1.5. The Y-axis represents the pathways, while the X-axis represents the gene ratio. The size of the bubbles indicates the number of differentially expressed genes enriched in each pathway, and the color of the bubbles reflects the significance of the enrichment.

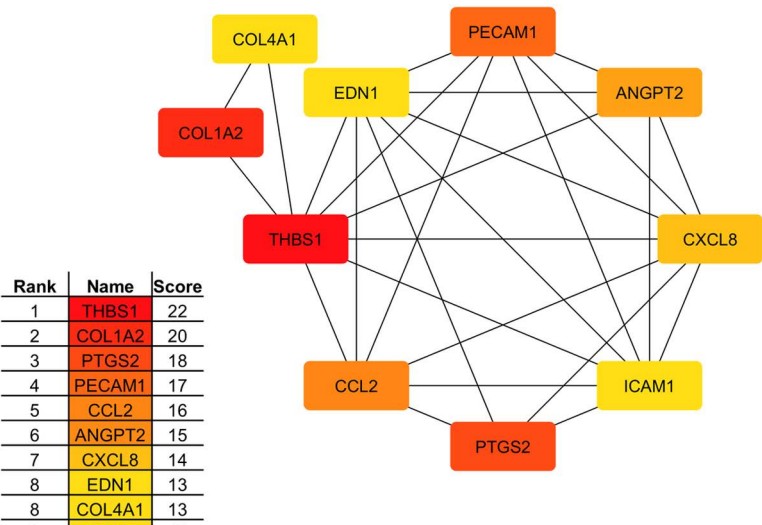

| Rank | Name | Score |
|------|------|-------|
| 1 | THBS1 | 22 |
| 2 | COL1A2 | 20 |
| 3 | PTGS2 | 18 |
| 4 | PECAM1 | 17 |
| 5 | CCL2 | 16 |
| 6 | ANGPT2 | 15 |
| 7 | CXCL8 | 14 |
| 8 | EDN1 | 13 |
| 8 | COL4A1 | 13 |
| 8 | ICAM1 | 13 |

**Fig 4. Top 10 hub genes identified by protein-protein interaction (PPI) network analysis from the differentially expressed genes (DEGs) between abscessed (*n* = 6) and healthy (*n* = 6) tissue livers from 12 beef × dairy steers.** Significant DEGs were defined as *adjusted p*-value < 0.05 and fold change > 1.5.

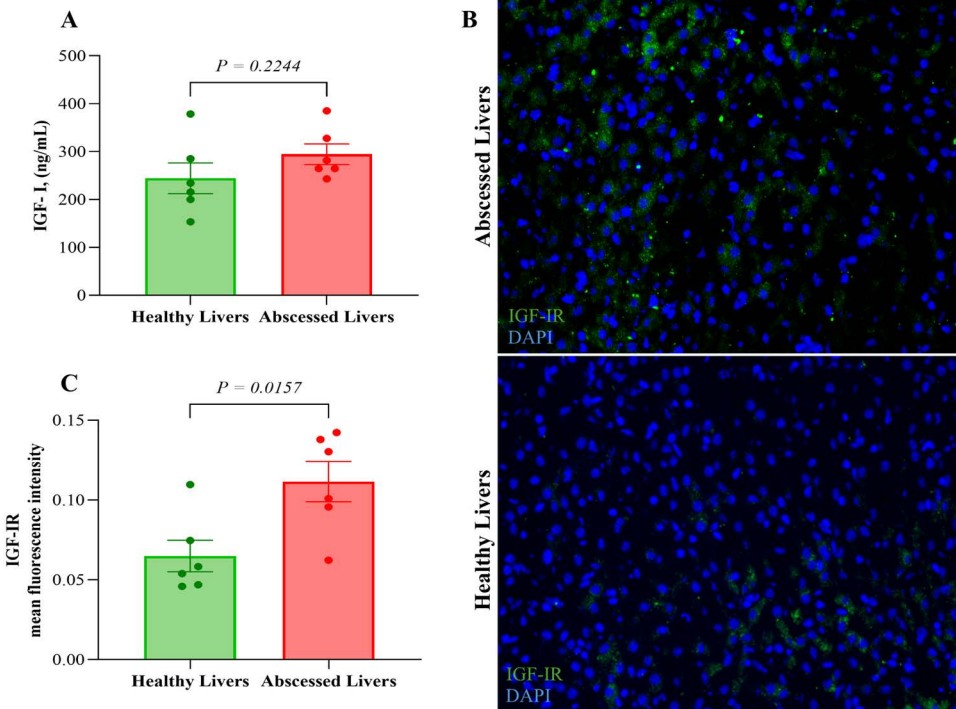

**Fig 5. Concentrations of IGF-I levels and immunohistochemistry of liver tissue samples from 12 beef × dairy steers, comparing abscessed (*n*=6) to healthy (*n*=6) livers.** A) Insulin-like growth factor-I (IGF-I) concentration in serum samples (*n*=12). The data represents mean IGF-I levels±SEM for each group (SEM=38.66). B) Immunofluorescence of abscessed (*n*=6) and healthy (*n*=6) livers in beef×dairy steers. Blue represents DAPI staining, while green indicates IGF-IR staining. C) Bar graphs represent the quantification of the mean fluorescence intensity of samples stained with IGF-IR antibody.

under pathological conditions, it can also be produced from other tissues, such as the heart, kidney, and liver [36–39]. A study demonstrated that during an inflammatory condition induced by lipopolysaccharides (LPS) in mice, Kupffer cells in the liver produce *FGF23* as part of the inflammatory response [40]. Considering that *F. necrophorum* is the primary etiological agent reported in liver abscesses in cattle, the up-regulation of *FGF23* in our study suggests that LPS from the bacteria could trigger higher *FGF23* production. This implies that the presence of *F. necrophorum* associated with LPS may initiate a cascade of inflammatory events in the liver, which may lead to the activation of Kupffer cells and subsequent production of *FGF23*.

Among the top up-regulated genes, neurexophilin 4 *(NXPH4)* is a neuropeptide-like glycoprotein from the NXPH family. *NXPH4* is an essential regulator of neuronal cells and plays a crucial role in intracellular signal transduction [41]. Recent studies demonstrated that elevated expression of *NXPH4* in human cancer cell lines and human serum is linked to hepatocellular carcinoma (HCC), suggesting this gene is a potential biomarker for the pathology [42,43]. Also, the same study with human cell lines observed that a higher expression of *NXPH4* is related to immune cell infiltration in HCC, indicating a role in tumor microenvironment modulation. This gene observed in our study may have an important role in immune cell infiltration into abscesses, potentially helping transport immune cells to the damaged area as a mechanism of the liver to re-establish the environment. However, further studies are needed to understand the *NXPH4* role in liver abscesses in cattle.

Cholesterol 7α-hydroxylase (*CYP7A1*) is a rate-limiting enzyme that catalyzes the first step of cholesterol into bile acids synthesis in the liver [44]. Abnormal bile acid metabolism and flow disruption in humans can lead to various hepatobiliary disorders. Biliary tract disease is the most common cause of human liver abscesses, accounting for 30% to 50% of cases

[45]. These abscesses often result from obstruction of bile ducts, which can lead to bacterial infection and subsequent abscess formation. While the direct link between *CYP7A1* expression and liver abscess formation in cattle has not been established, lower cholesterol circulation levels have already been reported to be linked to liver abscesses in cattle [11]. This suggests the hypothesis of a potential disruption in the liver biliary metabolism related to liver pathology.

Furthermore, following a PPI network analysis, the most significant genes identified were thrombospondin-1 (*THBS1*) and collagen type I alpha 2 chain (*COL1A2*). Many cells, including hepatocytes, endothelial cells, and stellate cells, produce THBS1 [46]. Under normal physiological conditions, the liver produces low concentrations of THBS1; however, in pathological conditions, such as alcohol cirrhosis, fibrosis, and nonalcoholic steatohepatitis, the up-regulation of *THBS1* was noted [47]. *COL1A2* is one of the principal components of the ECM and may play a role in encapsulating the abscesses as part of the fibrotic walls. In addition, a recent study reported that the *COL1A2* gene is associated with liver fibrosis in mouse models [48], which can be associated with excessive deposition of ECM.

GO analysis of DEGs from liver tissue, comparing abscessed to healthy livers, showed significant enrichment in GO terms. Some of these terms included angiogenesis, cell migration, cell adhesion, and endothelial growth factor (VEGF) receptor activity, which indicate that these biological processes play crucial roles in tissue repair and inflammation response. In addition, other significant terms involved ECM organization and extracellular region, which may suggest the deposition of ECM components, creating a scar in the liver tissue. Similarly, [49] described that liver pathologies were interconnected with angiogenesis, inflammation, and fibrogenesis. Furthermore, [15] reported that heifers with liver abscesses had lower white blood cells and lymphocyte concentrations in the blood, suggesting an immune response to the bacterial infection. Current results of the GO analysis may suggest a mechanism behind the progression of liver abscesses, which consists of the tissue trying to repair and regenerate while controlling inflammation.

Current assessment canonical pathway analysis revealed that DEGs were enriched in hepatic fibrosis signaling. This suggests an increased fibrotic response in the liver tissue, as an excessive accumulation of extracellular matrix leads to scar formation and impaired liver function [50,51]. Moreover, the hepatic fibrosis signaling pathway revealed several genes related to extracellular matrix formation, such as *BAMBI*, *COL1A2*, *CCN2*, *ITGA10*, *ITGA6*, and *TGFBR3*. Notably, *BAMBI* was down-regulated, and *TGFBR3* was up-regulated on this pathway. *BAMBI* acts as a negative regulator of the transforming growth factor-beta (TGF-β) as an antagonist to TGF-β1R [52]. On the other hand, the up-regulation of TGF-β receptors, including *TGFBR3,* is an important process due to the activation of hepatic stellate cells from the quiescent stage to myofibroblasts, which are the primary source of ECM accumulation [52]. Our findings are consistent with a recent study by [19], which also identified the activation of the hepatic fibrosis/ hepatic stellate cell pathway in the rumen of cattle with liver abscesses. This suggests that animals with liver abscesses trigger a fibrotic response, leading to increased ECM in the affected area, which could result in hepatic dysfunction.

The canonical pathway analysis identified the activation of interleukin signaling pathways involved in pro-inflammatory (IL-7, IL-8, IL-15, IL-17, and IL-33) and anti-inflammatory (IL-10 and IL-33) responses, as well as the activation of the NF-κB signaling pathway. The simultaneous activation of both pro-inflammatory and anti-inflammatory responses may suggest that the liver is attempting to control the bacterial infection through immune response mechanisms. In a study conducted in mice, which were inoculated with *Escherichia coli* to induce liver abscesses, the authors observed that mice that were resistant to liver abscess formation had reduced inflammatory cell recruitment and cytokine production, particularly those associated with LPS receptor TLR4 (Toll-like receptor 4) [53]. In our study, we did not find differences in the expression of TLR4; however, the activation of multiple pro-inflammatory pathways may contribute to the liver abscess formation. Furthermore, the activation of the NF-κB signaling pathway in the rumen comparing animals with and without liver absences has been reported [19], which aligns with our results, supporting current findings of inflammatory responses associated with liver abscesses.

Our analysis revealed activation of the insulin-like growth factor I (IGF-I) signaling pathway in steers with abscessed livers compared to healthy livers. This suggests a possible mechanism where the liver recognizes injury and attempts to

regenerate the affected tissue. Notably, we did not observe significant changes in circulating levels of IGF-I and the RNA-seq data for IGF-I expression when comparing animals with and without liver abscesses. These findings are consistent with [11], who also reported no differences in circulating IGF-I levels between beef cattle with and without liver abscesses. In addition, our study showed higher circulating levels of IGF-I in beef × dairy steers compared to the levels in straight-bred beef cattle found in other studies [9,10]. IGF-I levels in our animals were found to be in an average of 269.5 ng/mL for beef × dairy steers, independent of abscesses. This is consistent with a recent study reported by [9], which demonstrated that cattle with dairy influence have greater concentrations of IGF-I than straight beef cattle. Furthermore, steroid implants can also influence IGF-I circulation. [54] reported that implanted steers had increased serum levels of IGF-I and insulin-like growth factor binding proteins (IGFBP-3) compared to non-implanted steers. Our analysis also revealed that *IGFBP-1* and *IGFBP-3* genes were up-regulated among the DEGs, suggesting a mechanism for regulating the bioavailability of IGF-I in the liver. Notably, our RNA sequencing data and immunohistochemistry results revealed a novel finding that was a significant up-regulation of IGF-IR expression in the liver tissue of animals with abscesses. According to [55], healthy hepatocytes have low expression of IGF-IR; however, during early hepatocyte development or in hepatocyte injuries, the expression of IGF-IR increases. In addition, overexpression of IGF-IR in the liver has been related to hepatitis C, chronic hepatitis B, and cirrhosis, indicating a possible involvement of IGF-IR in liver damage [56]. Adding these findings together, although IGF-I levels did not increase in circulation, the liver attempts to regenerate the tissue by increasing the sensitivity with IGF-IR as a mechanism to maintain growth and metabolic functions in the body. On the other hand, if the up-regulation of IGF-IR persists, this activation may lead to uncontrolled cell proliferation in the liver.

## Conclusion

This study represents one of the first application of RNA-seq technology for liver transcriptomic analysis in beef × dairy steers comparing abscessed to healthy liver tissues. Our analysis has identified novel candidate genes and molecular pathways associated with liver abscesses in cattle, including inflammatory responses, tissue remodeling, and fibrosis. In addition, current research findings indicated that perturbations in the IGF-I signaling pathway in the liver are altered due to liver abscesses, while growth promotion technologies in cattle are known to increase both circulating and local IGF-I production. Further investigations are necessary to understand the complete mechanism behind liver abscess progression and IGF-I responses.

## Supporting information

**S1 Fig.  Insulin-like growth factor-I (IGF-I) concentration levels.**
(XLSX)

**S2 Fig.  Quantification of mean fluorescence intensity of IGF-IR–stained samples.**
(XLSX)

**S1 Table.  Differential expression genes (DEGs) comparing abscessed (n = 6) to healthy (n = 6) liver tissues from 12 beef × dairy steers.** Cutoff of adjusted p-value < 0.05 and fold change (FC) > 1.5.
(XLSX)

**S2 Table.  List of enriched Gene Ontology (GO) terms of differentially expressed genes (DEGs) comparing abscessed (n = 6) to healthy (n = 6) liver tissues from 12 beef × dairy steers.**
(XLSX)

**S3 Table.  List of canonical pathways analysis of differentially expressed genes (DEGs) comparing abscessed (n = 6) to healthy (n = 6) liver tissues from 12 beef × dairy steers.**
(XLSX)

**S4 Table. List of Protein-Protein Interaction (PPI) analysis of differentially expressed genes (DEGs) comparing abscessed (n = 6) to healthy (n = 6) liver tissues from 12 beef × dairy steers using STRING.**
(XLSX)

## Author contributions

**Conceptualization:** Bradley J Johnson, Luana D. Felizari.

**Data curation:** Dale R. Woerner.

**Formal analysis:** Luana D. Felizari, Jhones O. Sarturi.

**Investigation:** Bradley J Johnson, Sydney M. Bowman, Chiquito J. Crasto.

**Methodology:** Bradley J Johnson, Luana D. Felizari, Sydney M. Bowman.

**Project administration:** Bradley J Johnson.

**Supervision:** Bradley J Johnson.

**Validation:** Luana D. Felizari, Chiquito J. Crasto.

**Writing – original draft:** Luana D. Felizari.

**Writing – review & editing:** Bradley J Johnson, Luana D. Felizari, Jhones O. Sarturi.

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
