## [Decision Letter · Decision Letter 0]

17 Mar 2025

PONE-D-25-00830Hepatic transcriptome profiling unveils candidate genes in cattle with liver abscesses under the influence of beef genetics in dairy cattlePLOS ONE

Dear Dr. Johnson,

Thank you for submitting your manuscript to PLOS ONE. After careful consideration, we feel that it has merit but does not fully meet PLOS ONE’s publication criteria as it currently stands. Therefore, we invite you to submit a revised version of the manuscript that addresses the points raised during the review process.

We look forward to receiving your revised manuscript.

Kind regards,

Juan J Loor

Academic Editor

PLOS ONE

Reviewers' comments:

Reviewer's Responses to Questions

**Comments to the Author**

1. Is the manuscript technically sound, and do the data support the conclusions?

Reviewer #1: Yes

2. Has the statistical analysis been performed appropriately and rigorously? 

Reviewer #1: Yes

3. Have the authors made all data underlying the findings in their manuscript fully available?

Reviewer #1: Yes

4. Is the manuscript presented in an intelligible fashion and written in standard English?

Reviewer #1: Yes

5. Review Comments to the Author

Reviewer #1: General Comments

This paper describes an experiment designed to elucidate the mechanisms impacting the development and incidences of liver abscesses in crossbred cattle (Beef on Dairy). This is a very concerning problem for the industry and, as such, is an important paper for the beef and dairy industries. The paper is well prepared, and the approaches are sufficient to assess the question. The statistical approaches are appropriate, and the findings are consistent with the data presented.

Only some editorial and methodological clarifications are needed.

Line Comment

299 Here and throughout the manuscript and tables the authors use the work abscesses. However, the methods indicate that the hepatic tissue taken, was from the same region on the right side of the lobe. Were the samples from abscessed livers taken in the area of an abscess OR was the tissue sampled without regard for abscess presence in the tissue. In other words, did the sample “select” for abscessed tissue or did the tissue sample look normal? Regardless, “staining in abscessed livers, “ would read better.

327 This may be a journal style issue but, “Kumar et al. [39]

395 … responses is interpreted to suggest?

396 … conducted by Hulllahalli et al. …[52]

412 This comparison needs to be clarified. Higher than what?

435 Perturbations ?

Figures Can the quality of these images be improved. And Abscesses vs. Abscessed Livers

6. PLOS authors have the option to publish the peer review history of their article (what does this mean? ). If published, this will include your full peer review and any attached files.

**Do you want your identity to be public for this peer review?** For information about this choice, including consent withdrawal, please see our Privacy Policy .

Reviewer #1: No

---

## [Author Response · Author response to Decision Letter 1]

8 Apr 2025

299 Here and throughout the manuscript and tables the authors use the work abscesses. However, the methods indicate that the hepatic tissue taken, was from the same region on the right side of the lobe. Were the samples from abscessed livers taken in the area of an abscess OR was the tissue sampled without regard for abscess presence in the tissue. In other words, did the sample “select” for abscessed tissue or did the tissue sample look normal? Regardless, “staining in abscessed livers, “ would read better.

We thank the reviewer’s comment and apologize for the confusion. To clarify, we collected the same region from every animal without regard for the abscess presence in the tissue that was evaluated by scoring the whole liver. All the samples collected didn’t present any visible abscess, as the presence of an abscess could have introduced bacterial contamination, potentially affecting sequencing results. To make the text more comprehensive, we added a small statement in the method section to clarify that at line 106.

We also changed line 300, we changed it to: “staining in abscessed livers.”

327 This may be a journal style issue but, “Kumar et al. [39]

We agree this style doesn’t help the reader, so we changed it to: “A study……. [39]”.

395 … responses is interpreted to suggest?

We corrected the sentence now at line 396 with “…this may suggest…”

396 … conducted by Hulllahalli et al. …[52]

Also, here we updated the text, of this citation with: “In a study conducted in mice…[52]”.

412 This comparison needs to be clarified. Higher than what?

We thank the reviewer for the suggestion. We then changed the sentence in line 413 to make it more understandable for the reader as follows:” In addition, our study showed higher circulating levels of IGF-I in beef × dairy steers compared to the levels in straightbred beef cattle found in other studies [9, 10]. IGF-I levels in our animals were found to be…”

435 Perturbations ?

We corrected with perturbations.

Figures Can the quality of these images be improved. And Abscesses vs. Abscessed Livers

We thank the reviewer for the suggestion. We provided an updated version of the same representative images where a higher contrast of the Dapi and the IGF-IR staining will help to visualize better the differences between the groups. However, Figures 2, 3, and 4 made on R, it is not possible to increase their quality since we already saved them in higher quality as TIF File.

We apologize, and we agree that this can be confusing for the readers. We then modified the Figures (1 and 5) by changing the group abscesses to “Abscessed Livers” and healthy to “Healthy Livers”. We also changed the figure legends and the text accordingly.

---

## [Editor Report · Decision Letter 1]

22 Apr 2025

Hepatic transcriptome profiling unveils candidate genes in cattle with liver abscesses under the influence of beef genetics in dairy cattle

PONE-D-25-00830R1

Dear Dr. Johnson,

We’re pleased to inform you that your manuscript has been judged scientifically suitable for publication and will be formally accepted for publication once it meets all outstanding technical requirements.

Kind regards,

Juan J Loor

Academic Editor

PLOS ONE
---

## [Editor Report · Acceptance letter]

PONE-D-25-00830R1

PLOS ONE

Dear Dr. Johnson,

I'm pleased to inform you that your manuscript has been deemed suitable for publication in PLOS ONE. Congratulations! Your manuscript is now being handed over to our production team.

Kind regards,

on behalf of

Dr. Juan J Loor

Academic Editor

PLOS ONE